# Age-Related Changes in Functional Connectivity during the Sensorimotor Integration Detected by Artificial Neural Network

**DOI:** 10.3390/s22072537

**Published:** 2022-03-25

**Authors:** Elena N. Pitsik, Nikita S. Frolov, Natalia Shusharina, Alexander E. Hramov

**Affiliations:** 1Baltic Center for Artificial Intelligence and Neurotechnology, Immanuel Kant Baltic Federal University, Kaliningrad 236041, Russia; e.pitsik@innopolis.ru (E.N.P.); n.frolov@innopolis.ru (N.S.F.); nshusharina@kantiana.ru (N.S.); 2Neuroscience and Cognitive Technology Laboratory, Innopolis University, Kazan 420500, Russia

**Keywords:** functional connectivity, multilayer perceptron, aging, EEG, generalized synchronization

## Abstract

Large-scale functional connectivity is an important indicator of the brain’s normal functioning. The abnormalities in the connectivity pattern can be used as a diagnostic tool to detect various neurological disorders. The present paper describes the functional connectivity assessment based on artificial intelligence to reveal age-related changes in neural response in a simple motor execution task. Twenty subjects of two age groups performed repetitive motor tasks on command, while the whole-scalp EEG was recorded. We applied the model based on the feed-forward multilayer perceptron to detect functional relationships between five groups of sensors located over the frontal, parietal, left, right, and middle motor cortex. Functional dependence was evaluated with the predicted and original time series coefficient of determination. Then, we applied statistical analysis to highlight the significant features of the functional connectivity network assessed by our model. Our findings revealed the connectivity pattern is consistent with modern ideas of the healthy aging effect on neural activation. Elderly adults demonstrate a pronounced activation of the whole-brain theta-band network and decreased activation of frontal–parietal and motor areas of the mu-band. Between-subject analysis revealed a strengthening of inter-areal task-relevant links in elderly adults. These findings can be interpreted as an increased cognitive demand in elderly adults to perform simple motor tasks with the dominant hand, induced by age-related working memory decline.

## 1. Introduction

Functional connectivity (FC) of the brain network is defined as the presence of the temporal correlation between measures of activity recorded from the spatially distributed regions of the brain. Analyses of FC include a vast amount of metrics, most of them focusing on the synchronization of different brain rhythms associated with cognitive and motor-related activity [1,2,3,4]. The topology of such network changes depends on the type of human activity, and provides an assessment of the neural basis of cognitive load [5,6], working memory [7,8], and learning new skills [9,10]. Additionally, the structural changes in the functional connectivity pattern of the motor cortex imply motor learning [11] and is considered as an essential indicator for the development of brain–computer interfaces [12,13].

Based on this knowledge of the normal functional connectivity formation, this method is widely applied as a diagnostic tool for a variety of neurological disorders. There is evidence that the atypical functional connectivity network indicates autism spectrum disorder [14,15], Alzheimer’s disease [16,17], depression, and multiple sclerosis [18,19]. Moreover, the functional connectivity network is sensitive to age-related changes of the brain [20] and provides particularly valuable data on the physical and cognitive health in late life [21,22,23].

In the context of diagnostics, particular interest is dedicated to large-scale brain networks [24]. The interactions between remote brain regions reflect the neural response to human cognitive and motor-related activity [25]. For instance, the fronto–parietal area processes working memory, attention, and cognitive control [26,27,28,29]. Motor-related activity, including motor imagery, motor initiation, and planning, is associated with the sensorimotor system [30]. The disruptions of the normal large-scale functional connectivity pattern are often considered as a marker of neurological disorders [31,32].

For the purpose of evaluation of the large-scale brain networks’ interaction, we developed a method that provides an assessment of functional dependencies between the areas of the brain based on machine learning. To our knowledge, deep learning techniques are most widely used in EEG studies for the classification of various neurophysiological events. Particular interest is attached to one-dimensional convolutional neural networks (CNN-1D), which are naturally efficient with one-dimensional EEG time series [33,34]. However, in the present paper, we use the inherent ability of the deep learning methods to establish functional relationships between the sets of data with the goal to discover the coherence between EEG signals recorded from different areas of the brain. Depending on the goal of the study, the analysis of the brain activity can be conducted at the source or the sensor level. The source reconstruction analysis allows for bypassing the problem of field spread [35] by specifying the brain areas associated with the activity under consideration from the dynamics of the MEG or EEG time series [36]. In the present study, we analyze the activity of the motor-related activity of the brain areas as a joint activity recorded from several sensors, which allows us to reduce the effect of field spread and, therefore, conduct the analysis at the sensor level.

Our approach can be described in the terms of generalized synchronization, a type of synchronous behavior that indicates the presence of the functional relation between the drive and response systems and may be applied for the assessment of functional connectivity in neuroscience [37]. The main idea is that if the human motor action causes the emergence of functional relationships between different brain areas, then a pair of areas can be treated as two coupled systems: the drive system x(t) and the response system y(t) [38]:(1)y(t)=F(x(t)).

We propose detecting such a dependence using EEG recorded from different areas of the brain with the feed-forward multilayer perceptron (FF-MLP). The FF-MLP model is of particular interest in the current task, being known as a universal approximator with the ability to detect functional relationships between input and output data [39]. In our previous study, we used this approach to detect inter-areal functional connectivity in a young and healthy group of subjects during motor execution [40]. In the present research, we verify this method in the task of the age-related changes in brain functional connectivity associated with motor-related task performance. Our goal is to study the large-scale interactions between brain regions associated with motor execution and reveal the functional relations between them in two age groups of subjects. We expect to find significant differences in cortical activation at different stages of motor execution with a dominant hand, including the pre-movement period of a cued motor task. Our method revealed that two age groups induce different strategies of cortical activation, establishing different kinds of connections between remote brain areas in the mu- and theta-bands. We connect our findings with existing knowledge of healthy aging effects on brain activity response, which confirms the adequacy of the proposed method for assessing functional connectivity in neurophysiological data.

## 2. Materials and Methods

### 2.1. Experimental Dataset

Motor-related EEG data was recorded during experimental sessions with two groups of subjects: 10 healthy young individuals (YA group, 26.1±5.2 years, 3 females) and 10 healthy elderly subjects (EA group, 64.9±5.6, 4 females). Each subject gave informed consent to participate in the experiments. A detailed description of the experiment can be found in our previous work [41].

The experimental session began with a 5-min recording of the eyes-open resting state activity. Then, an active stage of the experiment began, consisting of repetitions of simple motor tasks on audio commands. All participants were instructed to clench their hand into a fist on the first audio command and to relax it after the second audio command of the same type. Two types of commands were presented: a short beep for the left hand (350 ms) and a long for the right (750 ms). Each participant performed a total of 30 movements with each hand. The duration of the period between two identical signals was chosen randomly in a range of 4–5 s. Between the motor tasks, a 6–8-s long pause was given before the next motor task. The timeline of the single motor task is shown in Figure 1A. During the session, we recorded an experimental protocol with the time marks corresponding to the each stimulus presentation. The timing was recorded both in absolute time, with the start of the experiment at 0 ms, and in computer time, in the format hh:mm:ss.sss. The latter was used to synchronise the beginning of the EEG data recording with the experiment start time. This protocol was used to identify the beginning of the each trial. In addition, the beginning of the motor execution was additionally detected using a surface electromyogram (sEMG) recorded from the long palmar muscle (see Figure 1C, upper row). Since our datasets consisted of right-handed subjects, we used the EEG data corresponding to the right hand movement only, expecting the most pronounced differences in cases with the dominant hand.

For brain activity data recording, we used an EEG amplifier Encephalan-EEGR-19/26 (Medicom MTD, Taganrog, Russia), which records EEG signals sampled at 250 Hz with 31 Ag/AgCl electrodes placed according to the “10–10” international system (see Figure 1B). A notch filter was applied to reject the 50-Hz component to avoid power-line interference. The oculomotor and cardiac artefacts were removed using independent component analysis (ICA). Additionally, we used the 5th-order Butterworth filter to obtain two subsets filtered in frequency bands of interest (FOI)—the theta- (4–8 Hz) and mu-bands (8–14 Hz), which are known to be associated with the processes of motor initiation and execution in healthy subjects; in our previous research, we revealed that theta-band activation is associated with age-related loss of the dominant hand functionality [41]. Finally, we inspected the dataset manually and removed the epochs with strong artefacts that could not be removed in the previous steps.

For the FF-MLP model, we segmented the EEG dataset into epochs centered at the moment of auditory stimulus presentation. Each epoch contained 1-s pre-stemulus recordings and 2-s post-stimulus recordings, which included sensorimotor integration and motor-related activity.

### 2.2. FF-MLP Model

#### 2.2.1. Training and Validation Datasets

We selected 15 sensors and divided them into 5 groups, corresponding to the brain areas of interest: the parietal area (*P*, sensors P4, Pz, P3), the frontal area (*F*, sensors F4, Fz, F3), the left hemisphere of the motor cortex ($MC_{L}$, sensors Fc3, C3, Cp3), the right hemisphere of the motor cortex ($MC_{R}$, sensors Fc4, C4, Cp4), and the midline of the motor cortex ($MC_{Z}$, sensors Fcz, Cz, Cpz). The corresponding areas are highlighted in Figure 1B. We chose the areas of interest based on existing knowledge of the brain cortex activation associated with sensorimotor integration [42]. For instance, the parietal cortex plays a key role in sensorimotor decision making and motor control [43,44]. The frontal cortex is known to be involved in the movement preparation process and in attention [45]. We also used sensors placed above right, left, and middle motor cortex, due to the contralateralization of the hand movements [46].

Each area of interest was represented by three sensors, based on the multivariate approach described in our previous work [47], according to which motor brain activity can be represented as a three-dimensional state trajectory, treating the constitutive time series as state variables. Consider each pair of the subsets x(t) and y(t) as:(2)$$\begin{array}{cc} x(t) & ={\left(x_{1}\left(t\right),x_{2}\left(t\right),x_{3}\left(t\right)\right)}^{T},\\ y(t) & ={\left(y_{1}\left(t\right),y_{2}\left(t\right),y_{3}\left(t\right)\right)}^{T}, \end{array}$$
where vectors x(t) and y(t) represent trajectories of electrical activity in the corresponding brain areas in a 3D state space formed naturally by the multivariate data (see Figure 1C). According to our pioneer study [37], functional connectivity can be tested as the quality of prediction of y(t), based on x(t). Inputs and outputs x(t) and y(t) were chosen without loss of generality.

#### 2.2.2. MLP Model Configuration

The proposed model establishes the functional relations between the sets x(t) and y(t) and their prediction $y^{\prime }\left(t\right)$. Therefore, a statistically significant approximation $y\left(t\right)≃y^{\prime }\left(t\right)$ indicates the presence of a functional connection between the brain areas corresponding to x(t) and y(t).

Our fully connected MLP model contains 3 input and 3 output linear units, according to the dimensionality of the multivariate datasets, and two hidden layers, each having 10 softmax units. The model was trained using an Adam optimizer with a learning rate of 0.001 for the 1000 iterations.

The goodness of fit between actual values y(t) and their predictions $y^{\prime }\left(t\right)$, i.e., a quantification of functional connectivity degree, is evaluated using the $R^{2}$-score:(3)$$R^{2}=1-\frac{\sum_{d=1}^{D}\sum_{i=1}^{N}{\left(y_{d}\left(t_{i}\right)-y_{d}^{\prime }\left(t_{i}\right)\right)}^{2}}{\sum_{d=1}^{D}\sum_{i=1}^{N}{\left(y_{d}\left(t_{i}\right)-{\bar{y}}_{d}\left(t\right)\right)}^{2}},$$
where D=3 is the output data dimensionality, $y_{d}^{\prime }\left(t\right)$ and $y_{d}\left(t\right)$ are the *d*th components of the predicted and actual vectors, and ${\bar{y}}_{d}\left(t\right)$ is the mean value of the actual time series $y_{d}\left(t\right)$.

The resulting 5×5 connectivity matrices, filled with the values of the pairwise $R^{2}$-score, were computed for each subject and each FOI within 5 time segments of data: (−1.0,0.0) s, (0.0,0.5) s, (0.5,1.0) s, (1.0,1.5) s, (1.5,2.0) s. Each pair of vectors x(t) and y(t) was chosen from the same period of time but from different brain areas. Thus, one obtains a time-resolved evolution of connectivity from prestimulus ((−1.0,0.0)-s time frame) to stimulus-related activity. A baseline correction of functional connectivity was applied by extracting the prestimulus connectivity from subsequent matrices.

In summary, the step-by-step process of training and validation sets’ generation goes as follows. At the first step, the whole EEG recording from each of the 18 selected channels was preprocessed using ICA and a notch filter to exclude the artefacts. Then, we extracted FOIs using the Butterworth filter in FOI (mu- and theta-bands) and obtained 30 motor execution epochs for each FOI. These sets of data were united into 5 groups according to the sensor’s position on the head surface. These groups were paired as the drive state x(t) and responce state y(t), and the purpose of the FF-MLP was to obtain an approximation $y^{\prime }\left(t\right)$ of the response state to assess the functional dependence between the original states. For the FF-MLP training, the EEG data was normalized in the range (0, 1) and randomly shuffled. The final dataset was divided equally into training and validation samples and fed to FF-MLP. The results of FF-MLP was a 3D approximation $y^{\prime }\left(t\right)$ of the original y(t), which were measured with the $R^{2}$-score to assess the quality of the approximation. This algorithm is illustrated in the block scheme in Figure 2.

### 2.3. Statistical Analysis

To reveal statistically significant differences in the brain functional connectivity on the different stages of motor execution for each FOI, we used the *t*-test with Bonferroni correction for multiple comparisons. Within-subject inference was performed via a paired *t*-test against the pre-stimulus values of $R^{2}$-score, while the between-subject inference was performed using an unpaired *t*-test.

## 3. Results

An example of the FF-MLP prediction of the response trajectory is shown in Figure 1D (right). For this example, we used an single EEG trial filtered in the theta-band corresponding to the pre-movement stage of motor execution. The $R^{2}$ for this example is 0.826, which indicates that a strong functional dependence between $MC_{Z}\left(t\right)$ and P(t) was revealed by the proposed model.

Next, we proceed to the analysis of statistical differences between the functional connectivity networks of remote brain areas and the different stages of motor execution. The results are shown in Figure 3. In the theta band, we observed significant growth of the functional dependencies between different areas of brain in the time intervals 1.5–2.0 s and 1.0–1.5 s in the YA and EA groups, respectively. Additionally, the EA group demonstrated an activation of theta-band connectivity between the left and midline motor cortex at the early stage of motor preparation (0.0–0.5 s after the audio command) that was absent in the YA group.

In the YA group, motor execution caused a statistically significant decrease in the connection between the frontal lobe and the right motor cortex in the mu-range, which continues to drop throughout the task (see Figure 3A, lower row). In the EA group, we observed the growth of the functional dependence between the right and middle motor cortex 1 s after the audio command, and a decrease in the frontal–parietal and parietal–middle motor cortex links after 1.5 s (Figure 3B, lower row).

Finally, we performed between-group comparisons of $R^{2}$-score adjacency matrices using an unpaired *t*-test. Results are shown in Figure 4. We demonstrated the strengthening of inter-areal task-relevant links in the EA group. In general, the couplings in the motor cortex are increased in the EA group in comparison with the YA group. In particular, the segments of the EEG associated with movement execution (1.5–2.0 s, see Figure 4) show the significant links between the frontal and right motor cortex both in the theta- and mu-bands. Moreover, in the pre-movement period (0.5–1.0 s), we observed a stronger functional dependence between the parietal and left motor cortex in the EA group in the mu-band. It should be noted that the EA group also demonstrated a stronger coupling between the parietal, frontal, and bilateral motor cortices.

## 4. Discussion

In the present paper, we applied the FF-MLP technique to EEG data with the aim to assess the motor-related functional connectivity patterns in two age groups. The restoration of the functional connectivity network is an important technique that allows connecting the mathematical toolbox with the biological bases of the subtle interactions between distinct brain areas. In addition, to the existing variety of mathematical methods for functional connectivity assessments from neurophysiological data, we propose using the inherent ability of machine learning methods to establish the functional relationship between two sets of data as an indicator of the functional connections between pairs of brain areas. Using a machine learning framework, we considered the age-related differences in motor-related functional connectivity of the brain network assessed via an artificial neural network. We found that the repetition of the motor tasks with the dominant hand under the audio command caused quite-distinctive patterns in the young and elderly adults groups. Overall, the EA group demonstrated a more pronounced pattern of functional connectivity at the different stages of motor execution, with theta-band activity playing a significant role in both motor preparation and motor execution.

We witnessed increased theta-band functional connectivity over the motor cortex in the EA group (Figure 1B) emerging right after the audio command (0.0–0.5 s). In our previous work [41], we demonstrated that elevated theta-oscillations in elderly adults are most probably explained by Bland’s sensorimotor integration model [48]. Contralateral theta activation in the primary motor cortex has recently been connected to increased motor learning [49]. Moreover, theta-band activity in the left dorsal premotor cortex has been linked to motor control and reprogramming [50]. It is possible that the age-related decline in brain plasticity leads to the broader activation of the sensorimotor cortex associated with motor planning in the elderly, while less neuronal resources are required for familiar motor task execution in their young counterparts. Supporting this conclusion, stronger activation in elderly adults was recently connected with compensatory neural processes [51], which suggests the involvement of more cognitive resources to maintain the same level of performance.

Despite the uncovered differences in functional connectivity, both groups demonstrated the statistically significant weakening of the mu-band relationships during motor execution. The EA group exhibits an increase in $MC_{Z}$–$MC_{R}$ connection, followed by a decrease in *F*–*P* and *P*–MCz links. The modulation of connectivity in the reported areas is consistent with the existing studies of motor action analysis [52]. In this meta-review, the authors showed that different domains of motor execution, including the action, preparation, and learning, involved the premotor, left, and right motor cortices, the supplementary motor area, and the parietal lobe. At the same time, a decreased functional connectivity in elderly adults was previously reported in [53], where the authors associated the decrease in post-training resting state functional connectivity with the temporary reduction in the between-areas’ connectivity in the motor network. At the same time, the premotor cortex activations are related to cognitive effort and perceptual decision making [54]. Therefore, the task-related modulation of the $MC_{Z}$–$MC_{R}$ connection and the following decrease in *F*–*P* coupling in elderly adults supports the previous assumption about age-related difficulties in new motor skills acquisition causing the observed activation pattern.

These conclusions were supported by the results of between-group comparisons (see Figure 4) that demonstrated the increased task-related activation in the EA group. In particular, we observed an increased theta-band coupling between the right and the midline motor cortices and mu-band coupling between the parietal and the left motor cortices. Such differences between the two age groups can be explained by the existing knowledge about are-related working memory decline [55]. We can conclude that in the EA group, there is most probably a low accessibility of the motor actions from memory [56,57], which slows down the motor-related brain activation and causes the predominance of sensorimotor integration processes aimed at the classification of the audio command and following motor planning.

## 5. Conclusions

In the present paper, we employed a ML-based approach for functional connectivity assessment to reveal the large-scale interactions between different brain regions based on the concept of generalized synchronization. We hypothesised that human motor action causes the occurrence of functional dependence between spatially separated brain areas that can be detected by a MLP-based algorithm. Our model demonstrated good ability to predict the states of the brain areas by the states of others, and allowed for reconstructing functional connectivity structures. We revealed that motor-related activity, including the pre-movement period, has quite-different patterns in the two age groups. Our findings suggest that in the elderly adult group, more resources are required to complete simple motor tasks on audio command, which is associated with higher functional dependence between remote brain areas on different stages of motor execution.

## Figures and Tables

**Figure 1 sensors-22-02537-f001:**
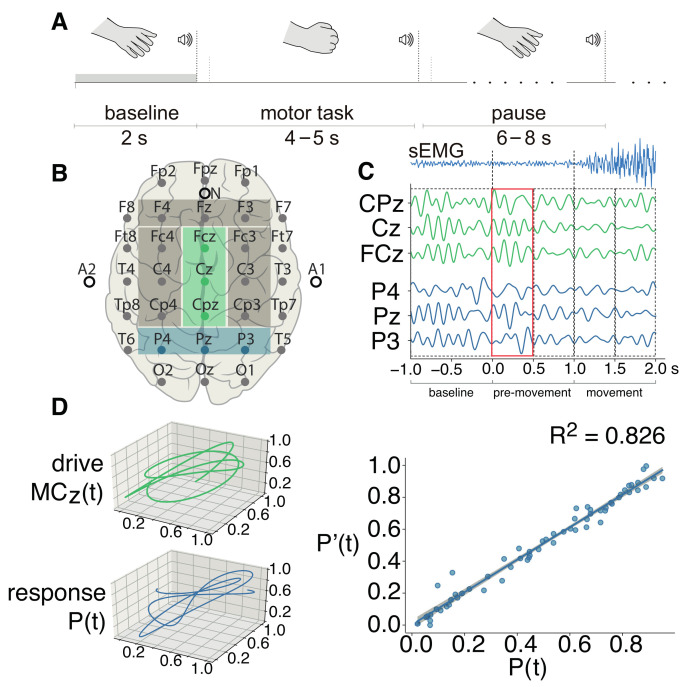
(**A**) The timeline of the single motor task; (**B**) the “10–10” international system of electrodes placement used in this study, with the areas of interest selected with colored rectangles. The green rectangle corresponds to the midline motor cortex (MCz={Cpz,Cz,FCz}), and the blue area highlights the parietal lobe (P={P4,Pz,P3}); (**C**) an example of EEG data filtered in the theta-band (4–8 Hz). Here, functional connectivity is computed between the brain areas MCz and *P*, based on multivariate EEG recordings. In the upper row, the sEMG averaged over the YA group is shown. The sEMG signal is filtered in the range of 10–100 Hz; (**C**) three-dimensional trajectories of subsets MCz and *P*; (**D**) the inference of functional dependence, where P(t) is the response state and $P^{\prime }\left(t\right)$ is the state predicted by the proposed FF-MLP model based on the drive state.

**Figure 2 sensors-22-02537-f002:**
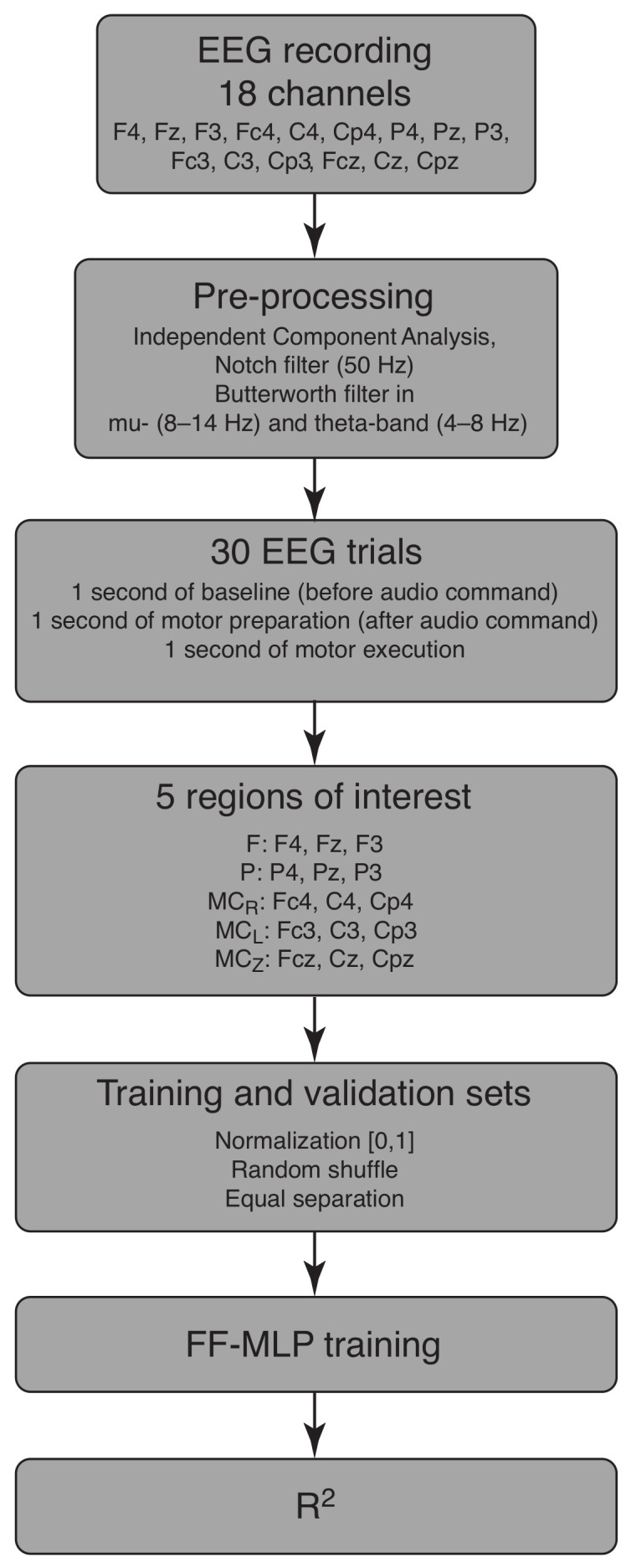
The block scheme of the research paradigm.

**Figure 3 sensors-22-02537-f003:**
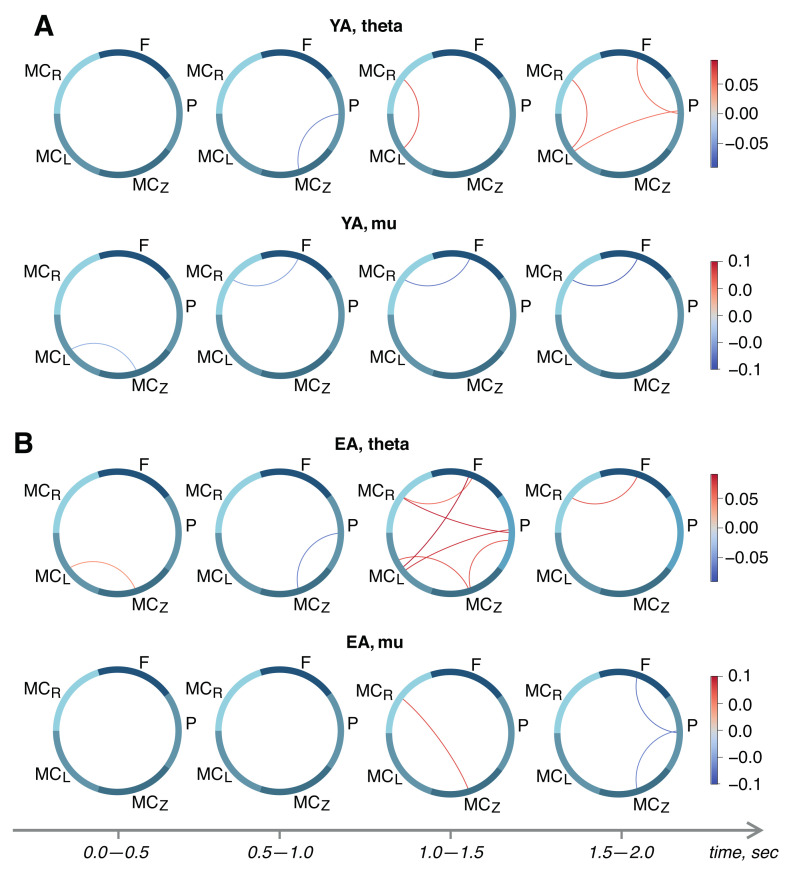
Within-subject differences between the $R^{2}$-score matrices in different time frames of stimulus-related activity for young adults (**A**) and elderly adults (**B**). Upper and lower rows in each subplot show connectivity in theta- and mu-bands, respectively.

**Figure 4 sensors-22-02537-f004:**
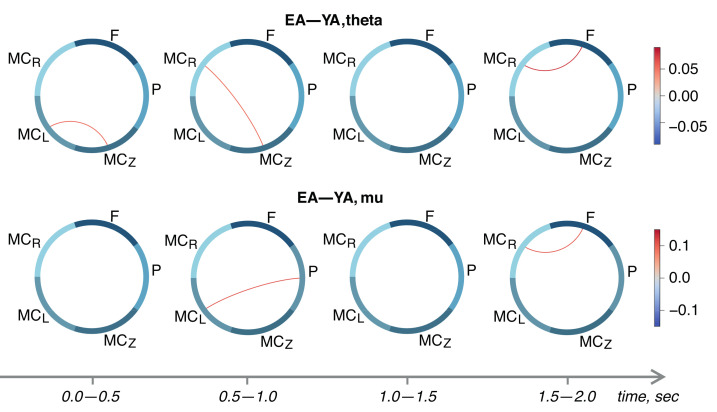
Between-group analysis of functional dependencies established by proposed FF-MLP model in the theta- (**upper** row) and mu-range (**lower** row).

## Data Availability

The datasets presented in this study can be found at dx.doi.org/10.608 4/m9.figshare.12292637.v2.

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
