# Peer review of "Age-Related Changes in Functional Connectivity during the Sensorimotor Integration Detected by Artificial Neural Network"

_sensors, 2022, doi:10.3390/s22072537_

Round 1
Reviewer 1 Report
This study proposes the brain connectivity to detect the age-related changes by using neural network. The manuscript is easy readable, but it can be benefited with more technical details, and after proofreading. The definition of functional connectivity should be better revised by the authors, as well as, the proposed methodology, especially the EEG pre-processing and the random process used to obtain both training and testing sets. I have some comments as follows:
Abstract:
1) The abbreviation AI should be defined. Please make a similar action with both YA and EA abbreviations on the line 143 pp. 4.
Introduction
2) The authors should review carefully the definition of brain areas/structures connectivity, EEG location connectivity, and functional connectivity. I think that this study analyzes the connectivity between EEG locations, instead of the functional connectivity. For instance, the functional connectivity helps to understand the role of specific brain areas given a task performance.
3) Theta band from 4 to 7 Hz is widely used in cognitive task studies, such as attention. Slow cortical potentials from 0.1 to 4 Hz, and event-related desynchronization/synchronization (ERD)/(ERS) from 8 to 30 Hz are widely used for motor task classification and analysis.
Materials and Methods
4) Use the term mu-band instead of alpha/mu band, as mu is widely employed in motor-related studies.
5) In section 2.1 Experimental dataset, the authors should explain better the trial structure (periods of each stimulus, period executing the motor task, period of the rest state, and pause). Maybe a figure can help to understand it. Specify if other signals, such as surface electromyography (sEMG) or other were further acquired to detect the movement onset, and accurately identify the motor preparation period and the movement execution period. It is one of my major concerns. The authors should describe the methodology used to identify these periods, useful to later analysis the connectivity between EEG locations.
6) Please check the frequency range for mu (8-12 Hz) and theta (4-7 Hz) for band-pass filtering. Also add a special filter, such as Common Average Reference (CAR) to reject physiological and non-physiological interference. It is other of my major concerns.
7) In Figure 1b, the authors should add the average of the muscular activation by using sEMG, identifying accurately the pre-movement and movement execution.
8) Here, it is another major concern. The FF-MLP training step should be revised and described in detail. Before any processing, I suggest to first select randomly the training and testing sets from 30 trials. After getting the training set, the authors can apply filters (CAR and band-pass) for pre-processing, EEG segmentation from -1 to 2 s, EEG normalization, selection of five areas of interest, and FF-MLP training. It is not clear if x(t) and y(t) in Eq (2) are selected from the same period (rest state, pre-movement, movement execution) during the FF-MLP training.
Results and Discussion
9) In Figure 2, the authors should add the average of the muscular activation by using sEMG.
10) Could the authors to verify or discuss the effect of auditory stimuli on the results?
11) The results should be better compared with the state-of-the-art, including especially studies using the brain area connectivity to investigate upper-limb movements. I also suggest reviewing these two references: Sosnik, Ronen, and Li Zheng. "Reconstruction of hand, elbow and shoulder actual and imagined trajectories in 3D space using EEG current source dipoles." Journal of Neural Engineering 18.5 (2021): 056011. Papitto, Giorgio, Angela D. Friederici, and Emiliano Zaccarella. "The topographical organization of motor processing: An ALE meta-analysis on six action domains and the relevance of Broca’s region." NeuroImage 206 (2020): 116321.
References
12) The references published 10 years ago can be removed or replaced.
Author Response
Abstract
Comment: The abbreviation AI should be defined. Please make a similar action with both YA and EA abbreviations on the line 143 pp. 4
Reply: Done.
Introduction
Comment: The authors should review carefully the definition of brain areas/structures connectivity, EEG location connectivity, and functional connectivity. I think that this study analyzes the connectivity between EEG locations, instead of the functional connectivity. For instance, the functional connectivity helps to understand the role of specific brain areas given a task performance.
Reply: In our understanding, the nature of the method used in this study is to find the functional relationships between the EEG signals recorded from the different brain area. Therefore, we discuss the results in the terms of functional connectivity, since we were able to assess the correlation between different brain areas activity.
Besides, depending on the goal of the study, the analysis of the brain activity can be conducted on source and sensor level. The source reconstruction analysis allows to bypass the problem of field spread [Schoffelen, J.M.; Gross, J. Human brain mapping (2009) 30] by specifying the brain areas associated with the activity under consideration from the dynamics of the MEG or EEG time series. However, our approach to analyze the activity of the brain areas as a joint activity recorded from several sensors allows us to reduce the effect of field spread and, therefore, conduct the analysis on the sensor level. We added this note to the Introduction section of the manuscript (lines 50-57).
Materials and Methods
Comment: Use the term mu-band instead of alpha/mu band, as mu is widely employed in motor-related studies.
Reply: Corrected.
Comment: In section 2.1 Experimental dataset, the authors should explain better the trial structure (periods of each stimulus, period executing the motor task, period of the rest state, and pause). Maybe a figure can help to understand it. Specify if other signals, such as surface electromyography (sEMG) or other were further acquired to detect the movement onset, and accurately identify the motor preparation period and the movement execution period. It is one of my major concerns. The authors should describe the methodology used to identify these periods, useful to later analysis the connectivity between EEG locations.
Reply: during the experiment, we recorded a protocol with the time marks corresponding to the each stimulus presentation. The timing was recorded both in absolute time (i.e. with the start of the experiment at the 0 ms) and computer time in format hh-mm-ss. The latter was used to synchronise the beginning of the EEG data recording with the experiment start. This protocol was used to identify the beginning of the each trial, which was checked during the pre-processing step. In particular, we applied the continuous wavelet transform and identified the movement onset as the first minimum of the event-related desynchronisation of each EEG trial, which also allowed us to select the time intervals corresponding to the pre-movement period. This method was described in detail in [Frolov et al PLOS ONE (2020) 15(9): e0233942], where the same time intervals were chosen using this approach.
Comment: Please check the frequency range for mu (8-14 Hz) and theta (4-7 Hz) for band-pass filtering. Also add a special filter, such as Common Average Reference (CAR) to reject physiological and non-physiological interference. It is other of my major concerns
Reply: In the present paper, we determine the frequency bands of interest with the accordance with the existing scientific evidence of how brain activity reacts on the motor-related events. For instance, mu-rhythm (8-14 Hz) is known to contribute in the motor execution, imagery and observation [Debnath R. et al NeuroImage (2019) 184], when theta-rhythm (4-8 Hz) activation is associated with the sensorimotor integration and reaction to the audio stimuli in elderly adults [Frolov et al PLOS ONE (2020) 15(9): e0233942].
As to the physiological and non-physiological interference, we performed artefact removal including undesired oculomotor and cardiac artefacts using the Independent Component Analysis (ICA) at the EEG pre-processing step. Parasitic oscillations of the electric field produced by the powerline at 50 Hz were removed via the Notch filter.
Comment: In Figure 1b, the authors should add the average of the muscular activation by using sEMG, identifying accurately the pre-movement and movevement execution.
Reply: We thank the referee for this important remark. The time intervals of motor-related brain activation, i.e., the motor-related desynchronization of the mu-band oscillations, were selected based on our previous study conducted on the same dataset [Frolov et al PLOS ONE (2020) 15(9): e0233942]. Besides, we checked the presence of muscular activity using sEMG and confirmed that the motor execution starts 1 second after the audio command, which allows us to suggest the motor preparation in the time segment [0,1] sec. We added the averaged sEMG on Fig. 1C.
Comment: Here, it is other major concern. The FF-MLP training step should be revised and described in detail. Before any processing, I suggest to first select randomly the training and testing sets from 30 trials. After getting the training set, the authors can apply filters (CAR and band-pass) for pre-processing, EEG segmentation from -1 to 2 s, EEG normalization, selection of five areas of interest, and FF-MLP training. It is not clear if x(t) and y(t) in Eq (2) are selected from a same period (rest state, pre-movement, movement execution) during the FF-MLP training.
Reply: The main purpose of FF-MLP is to approximate functional relation, if one is present, between the input X and the output Y. Otherwise, it fails to provide an accurate prediction. Given two time-series X = {x1,x2,…,xT} and Y={y1,y2,…,yT}, where xi and yi are the states of X and Y at the time moment i, we use FF-MLP to establish the presence of functional relationship between these processes, i.e., to find out if the trained FF-MLP can accurately map X to Y, in a trial-wise fashion. With this aim for each trial, we randomly shuffle the order of pairs (xi,yi), form the training and validation sets from the shuffled data, train the FF-MLP, and quantify its accuracy on the validation set using the R2-score. Higher R2-score indicates higher degree of functional connectivity between X and Y, and vice verse. This process is repeated for each trial. We illustrated the pre-processing steps as block-scheme and added it to the manuscript (Fig 2 in the revised version of the manuscript). For more details and justification of this approach on the simulated and ECoG data, please refer our previous study [Frolov et al CHAOS (2019) 29(9), 091101.].
Results and Discussion
Comment: Could the authors to verify or discuss the effect of auditory stimuli on the results?
Reply: The auditory stimuli differentially affects the theta-band cortical activation and functional connectivity associated with the process of sensorimotor integration. This issue has been extensively discussed in our previous work [Frolov et al PLOS ONE (2020) 15(9): e0233942], and lies beyond the scope of the current study. However, we added the brief discussion of this in the discussion section.
Comment: The results should be better compared with the state-of-the-art, including especially studies using the brain area connectivity to investigate upper-limb movements. I also suggest to review these two-references: Sosnik, Ronen, and Li Zheng. "Reconstruction of hand, elbow and shoulder actual and imagined trajectories in 3D space using EEG current source dipoles." Journal of Neural Engineering 18.5 (2021): 056011. Papitto, Giorgio, Angela D. Friederici, and Emiliano Zaccarella. "The topographical organization of motor processing: An ALE meta-analysis on six action domains and the relevance of Broca’s region." NeuroImage 206 (2020): 116321.
Reply: We thank the referee for the valuable references. We have discussed them in the revised manuscript (references 35 and 51)
References
Comment: The references published 10 years ago can be removed or replaced.
Reply: We replaced the references older that 10 years in the manuscript. However, some of the references are the fundamental pioneering studies, such as 38. Rulkov, N.F.; Sushchik, M.M.; Tsimring, L.S.; Abarbanel, H.D. Generalized synchronization of chaos in directionally coupled chaotic systems. Physical Review E 1995, 51, 980; and 35. Schoffelen, J.M.; Gross, J. Source connectivity analysis with MEG and EEG. Human brain mapping 2009, 30, 1857–1865.
Reviewer 2 Report
This paper showed an interesting application of predicting the states of the brain areas by the states of the others. However, there are some issues that are listed below.
Introduction
- The paragraph from line 41 to line 57 is very long, and the information in this paragraph is very important.
- In line 42, the reference [36] is very old, there are multiple recent methods that used the MLP model by involving new technologies such as CNN-1D. Also, there is a time gap between the author's recent work in reference [37], and reference [36].
Materials and Methods
- In line 60. It is recommended to use more subjects, in this regard, the author could use a recent machine learning scheme such as deep learning 1D to obtain more accurate results with diversity conditions.
- In line 80. The butter-worth filter was used as a feature extraction method to obtain FOI. However, it is recommended to use a smart updated filter such as a convolutional neural network (1D) to extract features that will be adaptive according to the condition (for instance, subject age) of the acquired signal.
FF-MLP model
- In line 114. The author does not explain why there are 3 inputs that are more related to the number of sensor output. Also, there are 3 outputs that indicate there are three different predictions for motor activation.
Results and Discussion
- In line 151. Did these results were obtained using filtered or unfiltered signals? “brain areas indicate the less effective use of cognitive resources in elderly adults.”. The authors might mention and explain the above.
Author Response
Introduction
Comment: In line 42, the reference [36] is very old, there are multiple recent methods that used the MLP model by involving new technologies such as CNN-1D. Also, there is a time gap between the author's recent work in reference [37], and reference [36].
Reply: This reference was replaced with the recent one 39. Taud, H.; Mas, J. Multilayer perceptron (MLP). In Geomatic approaches for modeling land change scenarios; Springer, 2018; pp. 451–455
Materials and Methods
Comment: In line 60. It is recommended to use more subjects, in this regard, the author could use a recent machine learning scheme such as deep learning 1D to obtain more accurate results with diversity conditions.
Reply: Post-hoc computation of the achieved statistical power given the sizes of considered groups (10 subjects per group, 20 total) and the alpha level of 0.05, the probability of correct rejection of the null hypothesis is 0.9960. Therefore, we find our sample size sufficient.
Comment: In line 80. The butter-worth filter was used as a feature extraction method to obtain FOI. However, it is recommended to use a smart updated filter such as a convolutional neural network (1D) to extract features that will be adaptive according to the condition (for instance, subject age) of the acquired signal.
Reply: We thank the referee for this suggestion, but the band-pass filtering exploited in this study (using the Butterworth filter or similar) is a standard procedure of extracting the brain oscillations of interest. We consider the suggested convolutional neural network as a potential feature extraction method in our next studies. Besides, the goal of FF-MLP in our research is not classification, but approximation: we use FF-MLP to approximate the activity from the one brain region by the activity of the another to assess the functional relations between them. We described this more thoroughly in the section «Methods» of the manuscript.
FF-MLP model
Comment: In line 114. The author does not explain why there are 3 inputs that are more related to the number of sensor output. Also, there are 3 outputs that indicate there are three different predictions for motor activation.
Reply: 3 outputs correspond to the each of the trajectory coordinates. Each set of data (trial, area) is a 3D-trajectory with the data from each of the 3 sensors covering the corresponding area acting as a set of coordinates. We described the multivariate approach for analyzing EEG data in [Pitsik et al. Chaos (2020) 30(2)].
Results and Discussion:
Comment: In line 151. Did these results were obtained using filtered or unfiltered signals? “brain areas indicate the less effective use of cognitive resources in elderly adults.”. The authors might mention and explain the above.
Reply: All results were obtained using the band-pass filtered data in the frequency bands of interest: theta- (4-8 Hz) and mu-band (8-14 Hz) oscillations. We added the note of this in the section ‘Methods'.
The interpretation of the results was rephrased and enforced with the literature. The updated description is presented in the section ‘Discussion’ of the revised version of the manuscript.
Reviewer 3 Report
- Citing of references, which are written in Russian, is not advised:
- Badarin, A.A.; Grubov, V.V.; Andreev, A.V.; Antipov, V.M.; Kurkin, S.A. Hemodynamic response in the motor cortex to execution of different types of movements. Izvestiya VUZ. Applied Nonlinear Dynamics 2022, 30, 96–108.
- A review of related work is absent.
- “Fig. 1 shows the step-by-step process of training and validation sets generation for FF-MLP model from EEG dataset.”. False. Such information is not provided in Fig. 1.
- “good approximation”. How to understand “good”? Is it correct to use the word “good”?
- Why “The final dataset was divided equally into training and validation samples.”?
Author Response
Comment: Citing of references, which are written in Russian, is not advised
Reply: We removed the Russian reference from the manuscript. We thank the reviewer for this comment.
Comment: A review of related work is absent.
Reply: We extended the Introduction and Discussion sections with more links on the potentially related works. Links [33,34] are related to the method under consideration. For the discussion, we added [49,50] explaining the theta-rhythm activation, [51] to interpret the stronger activation in elderly adults and [56,57] to discuss the between subject comparison results.
Comment: Fig. 1 shows the step-by-step process of training and validation sets generation for FF-MLP model from EEG dataset.”. False. Such information is not provided in Fig. 1.
Reply: We are grateful to the referee for this note. The wrong link was removed from the text. Instead, we added the block-scheme illustrating the pre-processing steps on Fig. 2.
Comment: “good approximation”. How to understand “good”? Is it correct to use the word “good’?
Reply: We thank the referee for this remark. We assessed the quality of approximation with R2-score and considered it as «good» after statistical evaluation. However, we rephrased «good approximation» in a more correct form «statistically significant», which means the presence of statistically significant differences between the achieved R2-score and baseline level.
Comment: Why “The final dataset was divided equally into training and validation samples.”?
Reply: We have the relatively small dataset in the present study. The split of the dataset in half provided the most accurate results of the approximation.
Round 2
Reviewer 1 Report
The revised manuscript is easy readable and provides enough technical details to facilitate its reproducibility. I would like to thank the authors for attending my comments and suggestions.